

# A High-Resolution Monitoring Approach of Canopy Urban Heat Island using Random Forest Model and Multi-platform Observations

Shihan Chen[1,2] , Yuanjian Yang [2*] , Fei Deng[1], Yanhao Zhang [2], Duanyang Liu[3,4], Chao Liu[2] and Zhiqiu Gao[2]

[1]School of Geodesy and Geomatics, Wuhan University, Wuhan 430079, China
[2]Collaborative Innovation Centre on Forecast and Evaluation of Meteorological Disasters, School of Atmospheric Physics, Nanjing University of Information Science & Technology, Nanjing 210044, China
[3]Key Laboratory of Transportation Meteorology, China Meteorological Administration, Nanjing 210008, China
[4]Nanjing Joint Institute For Atmospheric Sciences, Nanjing 210008, China

*Correspondence to*: Yuanjian Yang (yyj1985@nuist.edu.cn)

**Abstract** Due to rapid urbanization and intense human activities, the urban heat island (UHI) effect has become a more concerning climatic and environmental issue. A high spatial resolution canopy UHI monitoring method would help better understand the urban thermal environment. Taking the city of Nanjing in China as an example, we propose a method for

evaluating canopy UHI intensity (CUHII) at high resolution by using remote sensing data and machine learning with a Random Forest (RF) model. Firstly, the observed environmental parameters [e.g., surface albedo, land use/land cover, impervious surface, and anthropogenic heat flux (AHF)] around densely distributed meteorological stations were extracted from satellite images. These parameters were used as independent variables to construct an RF model for predicting air temperature. The correlation coefficient between the predicted and observed air temperature in the test set was 0.73, and the average root-mean-

square error was 0.72°C. Then, the spatial distribution of CUHII was evaluated at 30-m resolution based on the output of the RF model. We found that wind speed was negatively correlated with CUHII, and wind direction was strongly correlated with the CUHII offset direction. The CUHII reduced with the distance to the city center, due to the de-creasing proportion of built-up areas and reduced AHF in the same direction. The RF model framework developed for real-time monitoring and assessment of high-resolution CUHII provides scientific support for studying the changes and causes of CUHII, as well as the spatial

pattern of urban thermal environments.

## 1 Introduction

Throughout the world, cities have formed rapidly owing to population growth and people gathering in certain areas to settle and build their lives. Such urbanization brings not only economic development but also the Urban Heat Island (UHI) phenomenon (Oke, 1982; Grimmond and S.U.E., 2007; Mirzaei, 2015; Cao et al., 2016; Zhao et al., 2020). Two major types

of UHIs can be distinguished: (a) the canopy urban heat island (CUHI), and (b) the surface urban heat island (SUHI). The particular type of UHI is defined based on the height above the ground at which the phenomenon is observed and measured



(Oke, 1982). The UHI effect has become an indisputable fact and brings adverse impacts on urban ecology and energy consumption (Roth, 2007; Yang et al., 2019; Yang et al., 2020c; Zheng et al., 2020). UHIs amplify thermal stress, so people residing in urban areas are relatively more impacted during heatwave episodes (Koken et al., 2003; Estrada et al., 2017). A recent study on the global UHI predicted that about 30% of the world's population is exposed to lethal high temperatures for at least 20 days per year, and by 2100, this proportion was projected to reach 48% (Mora et al., 2017). UHIs also have the potential to impact vegetation phenology (Kabano et al., 2021), diurnal temperature range (Argüeso et al., 2014), water consumption and general thermal comfort (Salata et al., 2017). Due to its negative impacts, the UHI effect has become a key challenge in achieving urban sustainability and assessing this phenomenon has attracted increasing interest over the last decade or so (Corburn, 2009; Pandey et al., 2014; Malings et al., 2017). In general, both background weather conditions (e.g., the wind vector and heatwaves) and city-specific characteristics (including the presence of urban green space, properties of built-up materials, and the intensity of human activity) influence the UHI's mean intensity and variation (Zhao et al., 2014; Manoli et al., 2019). Concerning these factors, the UHI also shows significant intracity variability since urban areas are highly heterogeneous. Therefore, exploring the formation and causes of UHIs is crucial for decision-makers involved in the planning of urban developments and allocating public resources.

There are two main approaches to studying UHIs: numerical simulation and observation. Numerical simulation can reduce the need for a large number of observations and reveal mechanistic insights by investigating the impacts of cities on meteorological variables (Chun and Guldmann, 2014; Zou et al., 2014; Zhang et al., 2015; Taleghani, 2016; Li et al., 2020). For instance, (Zhang et al., 2015) investigated the influence of land use/land cover (LULC) and anthropogenic heat flux (AHF) on the structure of the urban boundary layer in the Pearl River Delta region, China, through a series of numerical experiments. However, it is important to acknowledge that numerical simulation is a simplification of the real world and cannot replace actual observations. Observational studies of UHIs are arguably more robust in their findings (Hu et al., 2016; Chakraborty and Lee, 2019; Dewan et al., 2021), and can mainly be categorized into the following three methods: (1) in-situ (field) measurement, (2) mobile measurements, and (3) remote sensing technology.

In-situ (field) measurements include meteorological measurements and high-density observations. It is easy to compare long-term series of air temperature (AT) between urban and rural stations based on meteorological observation data (Liu et al., 2006; Liu et al., 2008; Xinfa et al., 2008; Yang et al., 2012; Scott et al., 2018; Nganyiyimana et al., 2020). With the analysis of meteorological data in a long time series, the contribution and trend changes of UHI intensity (UHII) can be clearly discovered. Meanwhile, however, owing to the limitations of meteorological sites in terms of their spatial representation, it is difficult to build a comprehensive understanding on the spatial distribution of urban thermal environment parameters [such as urban canopy temperature, land surface temperature (LST) and vegetation] (Liu et al., 2008; Nganyiyimana et al., 2020). To overcome these limitations, high-density observation stations are used to explore the spatial distribution of the urban thermal environment and its relationship with the surrounding environment (Hu et al., 2016; Bassett et al., 2016; Ching et al., 2018; An et al., 2020). Deploying denser observation stations or urban microclimate surveys can to some extent compensate for the limitation of a coarse spatial resolution. However, such approaches are usually unsuitable for large-scale studies due to



restrictions imposed by certain natural conditions, social activities, as well as the high cost of construction and maintenance (An et al., 2020). For example, mobile transect surveys have been used in many studies (Merbitz et al., 2012; Akdemir and Tagarakis, 2014; Hankey and Marshall, 2015; László and Szegedi, 2015; Al-Ameri et al., 2016; Liu et al., 2017; Popovici et al., 2018), as they can easily obtain the distribution of parameters along a designed route using only a set of equipment attached

to a mobile vehicle. However, it is rather costly to obtain observations at a fine resolution, broad coverage and high synchronicity with such an approach.

To overcome these possible issues, LST data from aerial sensors and earth observing satellites are commonly employed in UHI studies, and so remote sensing data such as those from AVHRR (Roth et al., 1989; Caselles et al., 1991; Gallo et al., 1993a), Landsat (Y et al., 2007; Zhou et al., 2015; Zhao et al., 2016), MODIS (Peng et al., 2012; Zhou et al., 2015; Li et al.,

2017; Yang et al., 2018; Chakraborty and Lee, 2019), aerial images (Buyadi et al., 2013; Heusinkveld et al., 2014; Yu et al., 2020) and so on (Zhao et al., 2020; Gallo et al., 1993b; Qin et al., 2001; Chakraborty et al., 2020) are widely used to explain the spatial distribution of the surface UHI and its relationship with the local environment (e.g., LULC).Remote sensing data have good application prospects as they can provide fine resolution and wide data coverage at times when other ground-based observations cannot. However, due to the influence of precipitation and clouds, the retrieval of LST sometimes can be

challenging. In addition, each satellite remote sensing dataset has its own characteristics (Zhao et al., 2016; Chakraborty and Lee, 2019). For example, Landsat images have a high spatial resolution (30 m) that can show urban block sizes, but the temporal resolution is rather low (16 days). The MODIS LST dataset has the advantage of high temporal resolution (four times per day), but the spatial resolution is only 1 km (Yang et al., 2018). Meanwhile, the LST can only quantify the SUHI effect, which is seriously affected by meteorological factors, e.g., cloud and evaporation. In contrast, as an important indicator

reflecting the energy exchange between the atmosphere and land in the urban canopy, AT is more representative than LST. In particular, AT is more related with human health and ecological changes in cities (Ho et al., 2016).

UHI studies based on AT observed by meteorological sites suffer from limited spatial coverage, which impedes a comprehensive understanding on the influencing factors and causes of canopy UHI (CUHI). Thus, there is an urgent need to develop rapid, high spatiotemporal resolution AT and refined CUHI intensity (CUHII) estimation methods to explore the

mechanisms under which anthropogenic factors (e.g., urban land-use changes, anthropogenic heat emissions, urban morphology and size) and natural factors (e.g., meteorological conditions and geographical differences) influence the CUHIs of complex and diverse cities.

Therefore, in this study we: (1) based on remote sensing data, AT and wind speed data as well as other environmental information from meteorological observations, retrieved the AT data at a 30-m resolution in the study area by using machine

learning; (2) calculated the CUHII distribution based on the retrieved AT data, and further explored the shape, intensity and influencing factors of the UHI by combining local LULC, wind vector and urban morphology data.


## 2 Materials and Methods

### 2.1 Study areas

Nanjing, the capital city of Jiangsu Province in China, is located along the lower reaches of the Yangtze River and, as part of the Yangtze River Delta Urban Agglomeration, has a high level of urbanization. In fact, Nanjing has been experiencing rapid urbanization since China's economic reform in 1978. According to the National Bureau of Statistics, the population in Nanjing increased from 6.13 million in 2000 to 8.34 million in 2018. In 2016, the built-up area of Nanjing expanded to 773.79 km$^2$, pushing the city to rank as the ninth-largest among all Chinese cities (Wang et al., 2020b). The total GDP in 2020 was about 1.48 trillion CNY, ranking ninth among all Chinese cities. Along with economic development, Nanjing's UHII was observed to be 0.5°C in 2005 and is increasing at 0.109°C·(10 yr)$^{-1}$ (Xinfa et al., 2008).

### 2.2 Data

All of the satellite remote sensing data employed in this study are from the geospatial data cloud, including those gathered by the Landsat 8 Operational Land Imager (OLI). OLI has nine bands, including a coastal band, blue band, green band, red band, near-infrared band, two shortwave infrared bands, a panchromatic band, and a cirrus band. Due to the low temporal resolution (16 days) of the Landsat 8 OLI dataset and the vulnerability to cloud cover, data from three instances of cloudless conditions over Nanjing were selected for use in this paper—namely, 10:43 local time (LT) on 11 August 2013, 2 September 2015 and 21 July 2017. The specific band ranges and uses of Landsat 8 OLI are shown in Table S1 of the supplementary information. Meteorological observation data, including AT (0.5°C intervals on 11 August 2013 and 0.1°C intervals on 2 September 2015 and 21 July 2017), wind speed and wind direction, at 11:00 LT on the day closest to the satellite transit time, were selected. All weather stations in operation on those three days were included, numbering 218 totally and 63, 79 and 76 respectively (Figure 1). Figure 1 shows the 2-m AT and LULC on these three days. Compared with the LULC, the spatial patterns of AT on these three days are quite different (Figure 1).

In addition to global climate change, the influence of human activities on the CUHI cannot be ignored. Previous studies have pointed out that AHF is closely related to the change in built-up areas and population density around the stations, which reflects the fact that the effects from both anthropogenic emissions and land-use change are related to latent heat flux and sensible heat flux (Zhou et al., 2012; Yang et al., 2020b; Wang et al., 2020a; Zhang et al., 2021). Therefore, AHF was retrieved via a physical method (Bing Chen and Shi, 2012; Chen et al., 2012; Chen et al., 2014) based on 1000-m spatial resolution NOAA nighttime lighting data and with local economic development and energy consumption data, and the AHF data at the same time in Nanjing were provided by Chen et al (Bing Chen and Shi, 2012; Chen et al., 2012; Chen et al., 2014). And lastly, the digital elevation model (DEM) data (30-m spatial resolution) used in this study are based on the second version of ASTER-GDEM, which is provided by the Geospatial Data Cloud site, Computer Network Information Center, Chinese Academy of Sciences. (http://www.gscloud.cn)



## 3. Random Forest model framework for air temperature retrieval

### 3.1. Construction of Random Forest model

The Random Forest (RF) model is a highly flexible machine learning algorithm that can analyze data with missing values or noise and has good anti-interference ability. To date, the RF model has been widely used as a feature selection tool for high-dimensional data to, for example, identify the importance of variables and predict or classify related variables. In this study, an RF model was constructed for each time's dataset to evaluate the AT using the RF package in R language.

### 3.1.1 Data Preparation

The process of urbanization will have a significant impact on CUHIs (Zhou et al., 2015). To comprehensively take into account the local urban environment, 18 factors were selected as independent variables, including anthropogenic parameters (i.e., AHF), geometric parameters (distance from the city center, proportion of LULC area, altitude, longitude, latitude, slope, aspect), and physical parameters [proportion of impervious surface (IS) area, albedo, normalized difference vegetation index (NDVI), normalized difference built-up index (NDBI), green normalized difference vegetation index (gNDVI), soil-adjusted vegetation

index (SAVI), normalized difference moisture index (NDMI)]. The inversion methods for these environmental variables were as follows: Based on Landsat 8 OLI satellite data, the LULC in Nanjing was divided into four broad categories (built-up, cropland, vegetation, and water body) by combining a support vector machine method and visual interpretation. The remote sensing indices were calculated using corresponding bands (Yang et al., 2012; Shi et al., 2015). The IS and surface albedo data were extracted via multi-band information (Son et al., 2017; Liang, 2001). Then, the geometric center of the built-up area was

calculated as the city center, and the distances between the meteorological stations and the city center were calculated. Slope and aspect were calculated based on the DEM data using ArcMap 10.2. The methods used for extracting the IS data and calculating the remote sensing indices and surface albedo are given in Text S1, together with the accuracy of IS and albedo. All the above data (except for DEM, aspect and slope) were extracted for each of the three years corresponding to the three selected Landsat images. Taking the data on 21 July 2017 as an example, Figure 2 shows the spatial distribution of some of

the environmental parameters, i.e., IS, distance from city center, AHF, and NDVI, where high spatial consistency between these parameters and the urban structure can be seen. For example, high-density built-up areas correspond closely to high AHF and low vegetation cover.

Due to advection and turbulent transport, neighborhood surroundings can affect the local temperature (Yang et al., 2012; Shi et al., 2015). Therefore, a fixed buffer zone was built surrounding the meteorological stations. Within the buffer zone of each

station the proportion of IS area and that of each LULC type, and the average values of surface albedo, AHF, NDVI, NDBI, SAVI, gNDVI and NDMI were calculated. Together with longitude, latitude, altitude and distance to the city center, these parameters were fed into the RF model as independent variables, with AT as the target variable. In addition, to find out the optimal size of the buffer zones for the model, we compared the model performances for different buffer zone sizes, i.e., buffer





zones with a radius of 500 m, 1000 m, 2000 m and 5000 m, respectively. Figure 3 summarizes the research framework of this
paper.

### 3.1.2 Five-Fold Cross-Validation

This paper uses the coefficient of determination ($R^2$) and root-mean-square error (RMSE) as verification indicators. $R^2$ indicates
the degree of fit between the predicted AT and the observed AT, and the RMSE can reflect the credibility of the prediction
result.
The cross-validation (CV) method can be used to evaluate the performance of the RF model (Zheng et al., 2020). In this paper,
we employ the five-fold CV method, in which the entire dataset is randomly divided into five subsets—each time four subsets
are used to train the RF model, and the remaining one is used for validating. After constructing the model, the validation data
are used to calculate the current $R^2$ and RMSE, and the process is repeated until each of the five folds has been used as
validation data. The randomness in the process of selecting samples for modeling gives the model the advantage of being
robust and highly accurate. With enough decision trees, it can ensure that each sample is used as a training sample and a test
sample, effectively avoiding overfitting.

### 3.1.3  Variable selection and model  parameter setting

Since not every variable in the model makes a prominent contribution to the performance, deleting those variables that can
reduce the prediction accuracy can improve the performance and simplify the model. Therefore, the number of variables should
be minimized on the premise of improving or not affecting the performance of the model. The contribution of each variable is
judged by two indicators: the percentage increase in mean-square error (%IncMSE), and the percentage increase in node purity
(IncNodePurity). Using the backward selection method, the variable with the smallest contribution is identified and removed,
and the model is re-run. These steps are then repeated until only one variable remains. The $R^2$ and RMSE under different
combinations of variables were evaluated (Figure S1). For each model, the combination of variables with the largest $R^2$ and
smallest RMSE was selected. Using this approach, eight, seven and six variables were selected for the model on 11 August
2013, 2 September 2015 and 21 July 2017, respectively (Table 2).
To build an RF model, two important parameters need to be set: the number of decision trees (Ntree) and the number of
variables sampled at each node (Mtry). The RF models were established with Ntree from 50 to 1200, with 50 as the step length,
and Mtry from 1 to 16 respectively, with 1 as the step length to traverse all the parameters. Figure 4 presents the $R^2$ and RMSE
values in each five-fold CV test.
The principle of parameter selection is to choose a simpler model (smaller Ntree and Mtry) under the premise of good
performance. In the end, the optimal Mtry and Ntree based on the datasets on 11 August 2013, 2 September 2015 and 21 July
2017 were 7 and 200, 10 and 150, and 7 and 50, respectively.



## 3.2. Model testing

Table 1 compares the performance of the RF model with different buffer sizes (500 m, 1000 m, 2000 m, and 5000 m) in the five-fold CV. The RF model based on the dataset on 11 August 2013 and 2 September 2015 within 1-km buffer zones performed best, with an $R^2$ and RMSE of 0.57 and 0.65°C, and 0.59 and 0.69°C, respectively. On 21 July 2017, meanwhile, the $R^2$ and RMSE with a 2-km buffer zone were 0.47 and 0.80°C respectively, outperforming other buffer sizes. In this study, we chose the 2-km buffer radius. As can be seen from Table 1, on 11 August 2013 and 2 September 2015, the $R^2$ and RMSE

with the 1-km buffer zone was very close to that from the optimal buffer size, i.e., the 2-km buffer zone, whereas on 21 July 2017 the $R^2$ and RMSE with the 1-km buffer zone deteriorated considerably compared to that with the 2-km buffer zone. In addition, according to recent studies, the effective range that can influence local temperature is within 2 km (Yuyu Ren, 2011; Yang et al., 2012; Shi et al., 2015). Therefore, a 2000-m buffer was finally chosen in this study.

In addition, three methods of AT modelling were also compared—two linear regressions (stepwise linear regression (Alonso

and Renard, 2019; Mira et al., 2017) and geographically weighted regression (GWR) (Wang et al., 2020a; Li et al., 2021)) and one nonlinear regression (the RF model (Alonso and Renard, 2020)). A detailed description of the linear regression methods is provided in Text S2. Table 2 shows the performance of each model based on the dataset within a 2000-m buffer zone. Compared to the other methods, the RF model achieves better $R^2$ and RMSE, indicating its higher capability in fitting nonlinear and complex data and suitability for predicting AT (Zhu et al., 2019; Yoo et al., 2018).

## 3.3 Prediction accuracy of RF Models

Figure 5 compares the measured AT of the high-density automatic stations in the training set or testing set and the predicted AT of the RF model in the five-fold CV. In general, a large number of scattered points of predicted and observed AT are clustered around the 1:1 line, indicating good performance of the model. In the training set, the average $R^2$ and RMSE of the three models are 0.955 and 0.325°C respectively. The $R^2$ and RMSE using data on 11 August 2013, 2 September 2015 and 21

July 2017 are 0.948 and 0.295°C , 0.954 and 0.310°C, and 0.963 and 0.369°C, respectively, indicating high model accuracy. The result of the testing set shows that the average $R^2$ and RMSE are 0.535 and 0.719°C, respectively. Among them,  the prediction results achieved on 21 July 2017 are slightly less accurate than those obtained on the two other days. A smaller $R^2$ and larger RMSE were observed on 21 July 2017 (0.802, 0.468°C) compared to 11 August 2013 (0.655, 0.563°C) and 2 September 2015 (0.700, 0.574°C). Based on existing research (Oh et al., 2020; Venter et al., 2020) and follow-up discussion

(section 4.2.1), it can be concluded that the model performs best outside of the summer months, when the spatial variation in AT is low and wind velocities are high, corresponding to the model from 2 September 2015. In contrast, during the summer months, the performance of the model constructed with a high spatial variation of AT or low wind speed conditions decreases slightly, corresponding to the datasets on 21 July 2017 and 11 August 2013.

Furthermore, we used %IncMSE and IncNodePurity to determine the contribution of each variable (Table 3) and to compare

their importance. The NDVI, and the proportion of IS, vegetation and water body area all appeared in the three models,





indicating that vegetation, water bodies and human activities have important and universal impacts on the AT distribution. The distance to the city center appeared in the model based on the data on 2 September 2015 and 21 July 2017, and ranked high, implying the impact of urbanization on the heat island.

The absolute error for RF prediction is defined as difference in predicted AT and observed AT at each weather station(See

Figure S2). The relative error is defined as that absolute error divided by observed AT, which is shown in Figure 6. In general, the mean relative (absolute) errors by all stations are 0.07% (0.014°C), 0.04% (-0.025°C) and 0.05% (0.003°C) on 11 August 2013, 2 September 2015 and 21 July 2017, respectively. In detail, most of errors are concentrated between -0.49°C and 0.5°C over more than half of all stations for these three days (Figure S2), and more than 39.1 %/71.7%/86.3% of the total stations exhibit predictions with relative errors <1%/2%/3% (Figure 6), indicating good performance of RF models for most areas.


## 4. Refined CUHII assessment in Nanjing

### 4.1 Refined CUHII distribution

After establishing the model, a 2-km buffer area was created for each 30-m-resolution pixel and the same 18 independent variables were calculated. The constructed RF model took these pixel-wise variables as input and output AT for each pixel,

and hence we obtained the RF model–predicted AT map at 30-m resolution (Figure 7). CUHII is an important indicator to quantify the UHI effect, which is usually defined as the difference in AT at the same level between urban and rural areas [7,32], as follows:

$$\text{CUHII} = T - T_{\text{rural}}, \quad (3)$$

where $T$ is the predicted AT in each pixel and $T_{\text{rural}}$ is the average AT in the reference rural area. A square area of size 10 km

× 10 km was selected as the reference rural area in the northern part of Nanjing (Valmassoi and Keller, 2021). It was far from the city center and barely impacted by the UHI effect (Figure 7). The average AT in each reference rural area was 36.0℃, 27.8℃ and 34.7℃,respectively. Then, the CUHII distribution in Nanjing was calculated according to Eq. (3) (Figure 8).

Figure 7 shows that the AT on 11 August 2013 and 21 July 2017 was higher and that the AT ranges were 35.4–37.8°C and 33.6–36.4°C, respectively. The corresponding CUHII was strong, with more than 1.5°C in the downtown area (Figure 8). On

2 September 2015, the AT range was 26.8–29.1°C (Figure 7) and the CUHI was slightly weaker, with the maximum value at only 1.3°C (Figure 8). The three images from different seasons and different weather backgrounds led to significant differences in CUHII. On 2 September 2015, the overall CUHI was the weakest among the three days. Consistent with a previous study (Wang et al., 2020b), the summer CUHI in Nanjing was found to be generally stronger than that in autumn and winter. The difference between the maximal heat island and cold island intensity on 21 July 2017 was 2.8°C, the largest among the three

cases. Generally, the densely populated central city area has a large proportion of IS area, large anthropogenic heat emissions, and higher AT, showing an obvious UHI phenomenon (Figures 7 and 8). However, in urban areas with high vegetation





coverage or large water bodies, the AT decreases with weakened CUHII (Figures 7 and 8). The AT gradually decreases from the city center to the suburbs. Suburban areas, which are covered by more vegetation and water bodies, have significantly lower AT than central urban areas. At the boundary of the central city, high-AT areas and heat islands extend outward along
built-up areas and roads (Figures 7 and 8).

Against different weather backgrounds, the spatial distributions of AT and CUHII exhibit heterogeneity in urban Nanjing on different days. The high-AT area on 11 August 2013 extended from the city center to a wide range, and the extreme value of AT was the highest (Figure 7a), corresponding to the strongest CUHI (Figure 8a). Combined with Figure 1, we can see only a small range of vegetation coverage and water bodies in the central urban area, so the CUHII decreased slightly. Only in the
suburban water body and farmland areas were there large cold island areas. Nanjing is traversed by the Yangtze River. On 2 September 2015, the high-AT area was relatively small to the north of the Yangtze River. The AT on the Yangtze River was the lowest (Figure 7b), with the strongest cold island here (Figure 8b). The high-AT area extended from the central city to the south, and the cold islands in the southern water body and vegetation covered areas were not significant. On 21 July 2017, the distribution of the heat island was the opposite. There was a large area of high-AT to the north of the Yangtze River, and the
cooling effect of the Yangtze River was weak (Figure 7c). Meanwhile, the AT in the southern suburbs dropped significantly, and cold islands widely spread in water body and cropland areas (Figure 8c). Compared with the distribution of CUHII on 11 August 2013, the AT over the water bodies and hills in the northeast of the central city was lower, forming a large and strong cold island area.

To further explore the intensity and coverage of the CUHI on different days, the area (km$^2$) occupied by different levels of
CUHII on the three different days was calculated (Table 4). The CUHI area on 11 August 2013 accounted for 84.1%, whereas that on 2 September 2015 and 21 July 2017 only accounted for 80.2% and 81.2 %. Comparing the proportions by area of different CUHII levels, on 2 September 2015 the CUHII area at 0–0.5℃ accounted for 57.0%, concentrating in this range, while that at 1–1.5℃ was only 56.97 km$^2$, which was much lower than that of the other two days. The strongest cold island was lower than −1℃, and the overall CUHI effect was relatively weak. In contrast, on 11 August 2013, the area of the CUHII
in the range of 1–1.5°C and 1.5–2°C was 1486.89 km$^2$ and 82.96 km$^2$, respectively, which far exceeded that on the other two days. On 21 July 2017, the area where the CUHII was greater than 1.5°C was only 0.14 km$^2$. Therefore, the distributions of UHII on the three days were quite different and need to be discussed separately.

**4.2 Potential drivers of CUHII**

According to previous studies, three factors—the wind vector field (He, 2018), LULC (Cao et al., 2018; Wang et al., 2020b)
and the urban structure (Shahmohamadi et al., 2011; Li et al., 2020)—are the most important influencing factors of CUHIs. In this section, we explore these three drivers of CUHI in Nanjing city.





### 4.2.1 Relationship between CUHII and the wind vector field

The horizontal air flow has a significant impact on the intensity and shape of the CUHI (He et al., 2021). Figure 9 shows the wind vector field observed by weather stations on the three days analyzed in our study.

On 11 August 2013, the average wind speed at the stations was 0.70 m/s, most of which recorded calm wind (0–0.2 m/s) or soft wind (0.3–1.5 m/s) (Figure 9a). The main reason for this was that Nanjing was continuously controlled by the western Pacific subtropical high at this time and was therefore experiencing a continuous heat wave (Jing et al.)—conditions that are usually associated with low wind speeds, descending motion and stable weather, leading to increased CUHI strength (Figure 8a) (Wang et al., 2021). On 2 September 2015, the average wind speed was 1.53 m/s, which was a significant increase (Figure

9b). The overall northwesterly wind direction led to the CUHII being lower than that on 11 August 2013. Indeed, it has been noted in previous work that the wind direction will significantly affect the position and shape of a heat island (Bassett et al., 2016), and in the present study the northwesterly winds resulted in the CUHI extending from the built-up area to the southeast (Figure 8b) whilst weakening significantly in the northwest. On 21 July 2017, the average wind speed reached 3.07 m/s, with a southwesterly wind direction (Figure 9c). The CUHI effect weakened accordingly, extending to the northeast in the

downward wind, and the CUHI was significantly weakened in the southwest (Figure 8c).

On all three days the wind speed in the suburban areas was higher than that in the central city, and this is because there is no shelter provided by tall and dense buildings in the suburban areas, which is conducive to cooling from air convection and therefore a weakening of the CUHII (Yang et al., 2020a). That said, records show that, surprisingly, the boundary-layer mean wind speed in a city can be higher than its rural counterpart. On the one hand, Nanjing is traversed by the Yangtze River, and

the central city surrounds a large area of water, wherein the low surface roughness of the water is conducive to air convection. On the other hand, channeling/the Venturi effect might be an important factor. When the prevailing wind is parallel to the axis between buildings, it will be forced to enter between the buildings, resulting in higher wind pressure, which increases the wind speed (Droste et al., 2018).

In order to quantify the relationship, the average CUHII and standard deviation under different wind speeds at various

meteorological stations were calculated (Figure 10). On 11 August 2013, the maximum wind speed was 2 m/s, which bore no significant relationship with the CUHII (Figure 10a). On 2 September 2015 and 21 July 2017, the maximum wind speed reached 5 m/s and 6 m/s, respectively, which showed a significant negative correlation with the CUHI (Figures 10b and 10c). The greater the wind speed, the more significant the negative correlation.

There are two aspects concerning the influence of air convection on CUHIs. On the one hand, air convection will facilitate

horizontal advection cooling between urban and rural areas, thereby weakening the CUHI (T. Brandsma, 2003). The greater the wind speed, the more significant the cooling effect (Figure 10). On the other hand, horizontal convection transfers heat from the upwind to the downwind area, weakening the upwind CUHII and strengthening the downwind CUHII (Bassett et al., 2016) (Figures 8 and 9). Under different wind speeds, the synergy of these two aspects differs significantly. On 11 August 2013, the average wind speed was the smallest among the three days at only 0.7 m/s, and there was no uniform wind direction,





corresponding to the strongest CUHI. The distribution of the CUHI was highly correlated with that of built-up areas (Figures 8a and 9a). On 2 September 2015 the average wind speed was 1.53 m/s. Due to the combined effect of horizontal advection cooling and heat transfer, an upwind cold island appeared and, meanwhile, the downwind area received heat from the upwind area and the CUHII increased significantly (Figures 8b and 9b). On 21 July 2017 the average wind speed was 3.07 m/s, and the upwind CUHII also weakened (Figures 8c and 9c). Downwind, however, the urban heat convection was the dominant factor, which reduced the CUHII in some areas.

## 4.2.2 Relationship between CUHII and LULC

LULC also has a significant impact on CUHII (Wang et al., 2020). The average values and standard deviation of CUHII were calculated for each LULC type on the three days (Figure 11). On 11 August 2013, the CUHII in the built-up area was the strongest, exceeding 1.1℃, and in the water body areas it was the weakest at only 0.22℃ (Figure 11a). On 21 July 2017, the CUHII in the built-up area was the strongest at 0.62℃, and in the vegetation areas it was the weakest at 0.24℃ (Figure 11b). The CUHII on these two days was highest in the built-up area, followed by cropland, and then water bodies and vegetation. On 2 September 2015 the CUHII in the built-up area was the strongest at 0.32℃, while it was the weakest at −0.06℃ in the water body areas (Figure 11c).

Different LULC types have different effects on AT owing to their own intrinsic physical properties, mainly reflected in three aspects:

(1) Due to the good thermal conductivity and small specific heat capacity of the surface material in the built-up area, the ability to absorb shortwave radiation during the day is stronger than that of other land uses. The LST is significantly higher than that of the suburbs, and therefore the atmosphere is easily heated (Hong et al., 2018).

(2) Due to sufficient water availability in cropland and vegetation-covered areas, evaporation will increase the latent heat flux and cooling effect (Zhao et al., 2020; Zheng et al., 2018). In contrast, the surface humidity of the built-up area is low, with low corresponding latent heat flux. The difference in latent heat flux will increase the difference in AT between urban and rural areas. The latent heat flux of the water bodies is the largest, and the cooling effect is the most obvious.

(3) There is a significant correlation between LULC and wind speed (Chen et al., 2020). Areas with tall buildings in built-up areas have high surface roughness and low wind speed, whereas water bodies have low surface roughness and high wind speed. The surface roughness of vegetation-covered areas and cropland is somewhere between. The air convection will increase the sensible heat flux and reduce the AT (section 4.2.1). Therefore, LULC and air convection will jointly enhance or weaken the CUHII.

On 11 August 2013, the average wind speed and the difference in wind speed between different LULC types were small, and so was the difference in sensible heat flux. The difference in radiation and sensible heat flux was the main factor. On 21 July 2017, the average wind speed was the highest, and the synergy in the three aspects led to the CUHII over different LULC types being highest in the built-up area, followed by cropland, vegetation, and then water bodies. On 2 September 2015, the CUHII



was highest in the built-up areas, followed by vegetation, cropland, and then water bodies. This was due to the influence of low wind speeds, which would have produced heat transfer and made the CUHII shift from the built-up area to other LULC types (section 4.2.1).

### 4.2.3 Relationship between CUHI and urban structure

Human activities and urbanization have a significant impact on the spatial distribution of UHI (Shahmohamadi et al., 2011; Zhang et al., 2015; Li et al., 2020; Zong et al., 2021). To explore this influence, concentric rings with various radii (5 km, 10 km, 15 km, ..., 40 km) were created surrounding the city center. Within each ring, the average values and error ranges of AHF and CUHII, along with the average proportion of built-up area, were calculated. Figure 12 shows that the CUHII, AHF and proportion of built-up area all significantly decrease with increasing distance to the city center.

From a longitudinal perspective, the AHF and the proportion of built-up areas both increased year by year. The built-up areas of Nanjing on the three days were 982.78 km$^2$, 1076.19 km$^2$ and 1220.36 km$^2$, respectively. The proportion of built-up areas beyond 20 km to the city center increased, especially within the range of 20–25 km. The AHF also showed the same trend, which within the range of 20–25 km even exceeded that in the range of 15–20 km on 2 September 2015 and 21 July 2017. This shows that built-up areas and human influence were spreading from the city center to the surrounding areas during this period. However, the intensity and range of the CUHI did not increase with this trend, because the wind field and weather background have a stronger influence on CUHI than urbanization (Hong et al., 2018; Zong et al., 2021).

### 5. Discussion

Based on the RF model and combined with local environment and background weather data, the pattern and causes of CUHIs can be analyzed in detail. On 11 August 2013, Nanjing experienced a heat wave, with almost no horizontal convection of air (Figure 9a). In dry areas, such as built-up areas, the latent heat flux remained unchanged, but the high reflectivity of the surface raised the AT. In the heat wave period, the higher AT increased the latent heat flux in rural areas (Khan et al., 2020). For example, vegetation and water bodies alleviated the increase in AT in rural areas. This combined effect exacerbated the difference in AT between the urban and rural areas, making the overall CUHI the strongest (Nganyiyimana et al., 2020; Meili et al., 2021). In Figure 8a, it can be seen that the cooling efficiency of vegetation in the urban area was not high and the coverage of the cooling area was small. This is because the stomata of leaves would have been closed under high AT and dry weather, resulting in reduced evapotranspiration and increased AT (Manoli et al., 2019). On 2 September 2015, northwesterly winds prevailed (Figure 9b), and there was abundant water vapor over the hills of northeast Nanjing and over the Yangtze River. The increase in latent heat flux and horizontal convection cooling lowered the CUHII. Cold islands even appeared to the north of the Yangtze River. The CUHII in the southeast direction was strong (Figure 8b), which was mainly affected by the heat transport of the prevailing winds (Chuanyan et al., 2005), causing the CUHI to shift toward the downwind area. On 21 July 2017, southwesterly winds prevailed in Nanjing, with high wind speed, decreasing the CUHII in the upwind region



(Figures 8c and 9c). However, there were large areas of vegetation coverage in the range of 10–20 km in the downwind region, which was affected by the combined effects of land use and horizontal advection cooling leading to lower CUHII than that of

20–30 km. This also confirms the conclusion (Bassett et al., 2016) that the upwind horizontal advection cooling has the strongest correlation with the weakening of the CUHI effect, and that the downwind region is affected by the wind speed.

There are four main methods for retrieving AT for CUHII assessment:

(1) Statistical methods (Prihodko and Goward, 1997; Alonso and Renard, 2020; Li et al., 2021). Statistical models of environmental factors and temperature are established to evaluate the AT, such as multiple linear regression models, partial

least-squares regression and GWR. In  previous study (Alonso and Renard, 2020), two methods of AT prediction (namely, stepwise linear regression and GWR) were compared with the RF model. The RF model has the highest accuracy and effectively avoids the problem of autocorrelation by filtering variables, which is consistent with previous work (Yoo et al., 2018; Zhu et al., 2019) and our present work. While conventional statistical methods, in addition, cannot effectively solve nonlinear problems (Oh et al., 2020).

(2) Temperature–vegetation index method (VTX) (Goward, 1997; Stisen et al., 2007; Vancutsem et al., 2010). This refers to inversion using the relationship between AT, LST and vegetation index under the premise that the temperature of a dense vegetation canopy is similar to the AT. While VTX only indicates the relationships between underlying surface, LST and AT. In fact, there are many factors that can affect AT, e.g., anthropogenic heat, altitude, and distance to city. Ignoring these factors, therefore, the accuracy of VTX method was low (Stisen et al., 2007)). In contrast, our RF model input multi-variables,

including more affecting AT factors.

(3) Machine learning methods (Venter et al., 2020). Predictions are made by establishing models of various variables and AT, such as RF models or neural networks. Compared with other machine learning methods such as neural networks (Astsatryan et al., 2021), the RF model has better noise immunity and is suitable for small sample sizes in this study. Other machine learning methods usually require a lot of data with little noise, so the data cleaning before modeling will take more time.

(4) Physical model methods. This category mainly constitutes the energy balance method (Yang et al., 2018), which refers to the study of AT inversion using the principle of energy balance. Physical model approach is relatively complex, and the performance is highly dependent on the understanding of the mechanism affecting AT, which can only address specific problems. While the RF framework in this paper is relatively simple, comprehensive, and suitable for different weather backgrounds.


Moreover, the RF prediction framework proposed in this work can not only dynamically predict CUHII in detail within highly heterogeneous cities, but can also be built against different weather backgrounds, mainly because the environmental parameters entered into the model are relatively stable within a certain period (such as the same month or season). As long as the environmental parameters are acquired once, they can be combined with the AT data in real time to establish the RF model,

and the spatial distribution characteristics of CUHII with high temporal and spatial resolution can be obtained. For instance, we randomly predicted the 30-m-resolution AT and spatial distribution of CUHII (Figure 13) with the wind vector field (Figure





S3) during the heat wave period of 12–14 August 2012, thereby supporting those involved in making decisions with respect to urban climate, urban planning and urban energy consumption. It is worth mentioning that there are many factors that affect the CUHII. In addition to human activities and LULC, the background weather conditions (such as heat waves, air pollution,

atmospheric circulation and cloud cover) are also extremely important (Bassett et al., 2016; Yang et al., 2020a; Khan et al., 2020), which should be introduced to improve the RF model of CUHII.

## 6. Conclusions

Taking Nanjing as an example and using remote sensing data with data from local weather stations, parameters to characterize the urban environment were constructed—namely, anthropogenic parameters (i.e., AHF), geometric parameters (distance from

city center, proportions of LULC types by area, altitude, and latitude and longitude, slope and aspect), and physical parameters (proportion of IS, surface albedo, NDVI, NDBI, SAVI, gNDVI and NDMI). A 2-km buffer zone was created around the meteorological stations, and the observed environmental parameters were extracted. A refined assessment framework of CUHII was then established by using Random Forest Model with observed AT and environmental variables.

Results showed that the correlation coefficient between the predicted and observed AT was 0.731, and the average RMSE was

0.719°C, indicating accurate results. Finally, the high-spatial-resolution (30m) CUHII distribution was analyzed. It was found that the shape of the CUHII was highly correlated with the spatial distribution of AHF and built-up area under calm wind conditions. Under the prevailing wind conditions, the CUHII should be discussed separately in upwind and downwind areas divided by the central city. In the upwind area, there was a significant negative correlation between the wind speed and CUHII. The higher the wind speed, the more obvious the negative correlation. In the downwind area, horizontal convection cooling

was found to be the leading factor under high wind speed weather, and heat transfer was the leading factor under low wind speed weather. The combined effects of built-up areas, heatwaves, and human factors can strengthen the CUHII, while the vegetation canopy and water bodies will weaken it. Vegetation and water bodies in the central urban area were found to have a significant cooling effect, providing a reference for urban development. With increasing distance from the city center, the CUHII decreased sharply.

In general, overlapping the refined CUHII with local environmental variables and weather conditions helps to explore the causes of CUHIs in more detail, instead of being limited to the location of meteorological sites and frequent changes in various types of weather. The new 30-m resolution CUHII evaluation framework developed in this study has strong portability, and important practical value. Our findings are helpful towards improving our understanding of the relationship between human activities and regional climate change, which can provide important guidance for urban development planning and allocation

of public resources in the context of global warming and rapid urbanization.

**Code availability**

The model in this paper is based on the random Forest data package in the R language, and our implementation and analysis code are available upon request to the corresponding author (yyj1985@nuist.edu.cn).

**Data Availability**

Landsat 8 OLI datasets (http://www.gscloud.cn/sources/index?pid=263&rootid=1&label=Landsat8&sort=priority&page=1 (access on 10 April 2021)) were used to retrieve IS area, LULC and NDVI. nighttime light satellite datasets (http://ngdc.noaa.gov/eog/dmsp/downloadV4composites.html (access on 10 April 2021)) were used to retrieve anthropogenic heat flux (AHF), surface meteorological observations were collected from the China Meteorological Data Service Center (http://data.cma.cn/en(access on 10 April 2021)), DEM were achieved from geospatial data

**Supplement**

The following are available online, Text S1: Specific inversion steps of related environmental variables. Table S1: band ranges and the main use of Landsat 8 OLI. Figures S1: The performance of the RF models under different variable combinations. Figure S2. the predicted error of the air temperature by random forest: (a) 11 August 2013; (b) 2 September 2015; (c) 21 July 2017. Figure S3: Spatial distribution of air temperature and wind vector field in Nanjing and the reference rural area during
12-14 August, 2013.

**Author contributions**

Conceptualization, Y.Y.; methodology, S.C. and Y.Y.; Data Curation: D.L. and S.C.; software, S.C.; validation, S.C. and Y.Y.; formal analysis, Y.Z., S.C., D.L., C.L. and Y.Y.; writing—original draft preparation, S.C.; writing—review and editing, F.D, S.C., Y.Z., C.L., D.L. and Y.Y.; visualization, S.C. and Y.Z.; supervision, Y.Y. and F.D. ; funding acquisition, Y.Y. All
authors have read and agreed to the published version of the manuscript.

**Competing interests**

The authors declare that they have no conflict of interest.



**Financial support**

This research was supported by the National Key Research and Development Program of China (grant no. 2018YFC1506502),
the National Natural Science Foundation of China (grant no. 42175098 and 42061134009), and the University Student
Innovation Training Project of Nanjing University of Information Science and Technology (grant no. 201910300283).

**Acknowledgements**

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



**Table 1: $R^2$ and RMSE of the RF model with different buffer sizes (500 m, 1000 m, 2000 m, 5000 m).**

|  | 500 m | | 1000 m | | 2000 m | | 5000 m | |
|---|---|---|---|---|---|---|---|---|
|  | $R^2$ | RMSE (°C) | $R^2$ | RMSE (°C) | $R^2$ | RMSE (°C) | $R^2$ | RMSE (°C) |
| 11/08/2013 | 0.33 | 0.75 | **0.57** | **0.65** | 0.56 | 0.65 | 0.36 | 0.74 |
| 02/09/2015 | 0.58 | 0.70 | **0.59** | **0.69** | 0.57 | 0.70 | 0.49 | 0.76 |
| 21/07/2017 | 0.19 | 0.92 | 0.17 | 0.91 | **0.47** | **0.80** | 0.16 | 0.93 |



**Table 2: $R^2$ and RMSE of stepwise regression, GWR (geographically weighted regression) and the RF model within a 2-km buffer zone.**


|  | Stepwise regression | | GWR | | RF model | |
|---|---|---|---|---|---|---|
|  | $R^2$ | RMSE (°C) | $R^2$ | RMSE(°C) | $R^2$ | RMSE (°C) |
| 11/08/2013 | 0.30 | 0.69 | 0.33 | 0.77 | **0.56** | **0.65** |
| 02/09/2015 | 0.47 | 0.74 | 0.44 | 0.82 | **0.57** | **0.70** |
| 21/07/2017 | 0.27 | 0.90 | 0.12 | 0.93 | **0.47** | **0.80** |





**Table 3. Importance of input variables for the RF model of AT estimation on the three different days.**

| 11/08/2013 | %IncMSE | IncNodePurity |
|---|---|---|
| Water-body | 9.23 | 4.71 |
| NDVI | 8.38 | 4.22 |
| NDBI | 7.15 | 7.13 |
| IS | 6.93 | 6.46 |
| Built-up | 4.19 | 2.05 |
| Vegetation | 2.35 | 1.38 |
| AHF | 0.89 | 2.91 |
| Cropland | 0.27 | 1.70 |

| 02/09/2015 | %IncMSE | IncNodePurity |
|---|---|---|
| Cropland | 5.10 | 9.45 |
| Distance to city center | 4.57 | 8.59 |
| Water body | 4.00 | 11.05 |
| NDVI | 3.18 | 5.34 |
| NDBI | 2.44 | 4.05 |
| Built-up | 2.41 | 2.78 |
| SAVI | 1.49 | 2.67 |
| Vegetation | 1.44 | 2.24 |
| IS | 0.40 | 2.59 |


| 21/07/2017 | %IncMSE | IncNodePurity |
|---|---|---|
| Distance to city center | 20.01 | 16.22 |
| IS | 18.36 | 15.75 |
| Vegetation | 11.52 | 8.08 |
| NDVI | 9.89 | 3.85 |
| gNDVI | 7.86 | 3.28 |
| SAVI | 6.78 | 2.38 |
| Water-body | 6.45 | 6.24 |

**Notes: NDVI, normalized difference vegetation index; IS, impervious surface; AHF, anthropogenic heat flux; DEM, digital elevation model; NDBI , normalized difference built-up index; gNDVI, green normalized difference vegetation index; SAVI, soil-adjusted vegetation index.**





**Table 4. Area occupied by different levels of urban heat island intensity on different days (km².**

|  | −1.5 to −1 | −1 to −0.5 | −0.5 to 0 | 0 to 0.5 | 0.5 to 1 | 1 to 1.5 | 1.5 to 2 |
|---|---|---|---|---|---|---|---|
| 11/08/2013 | 0.00 | 0.15 | 1047.43 | 1517.19 | 2446.03 | 1486.89 | 82.96 |
| 02/09/2015 | 0.02 | 192.13 | 1109.89 | 3751.88 | 1472.26 | 56.97 | 0.00 |
| 21/07/2017 | 0.23 | 232.52 | 1005.04 | 2670.11 | 2040.98 | 634.13 | 0.14 |

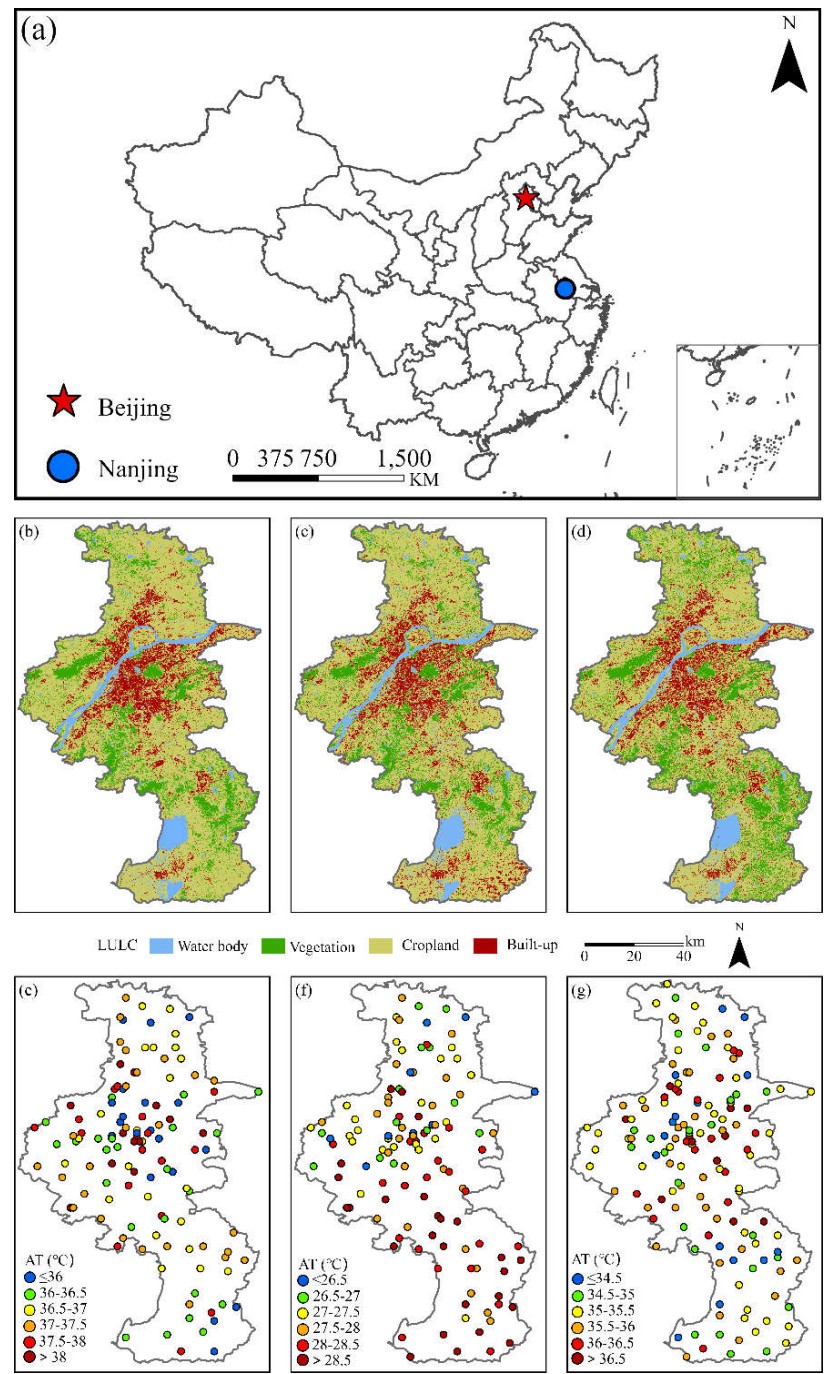

**Figure 1: Land use/land cover of Nanjing city and locations of automatic meteorological stations in Nanjing with recorded air temperature: (a) location map of Nanjing in China; (b, e) 11:00 local time (LT) 11 August 2013; (c, f) 11:00 LT 2 September 2015; (d, g) 11:00 LT 21 July 2017.**




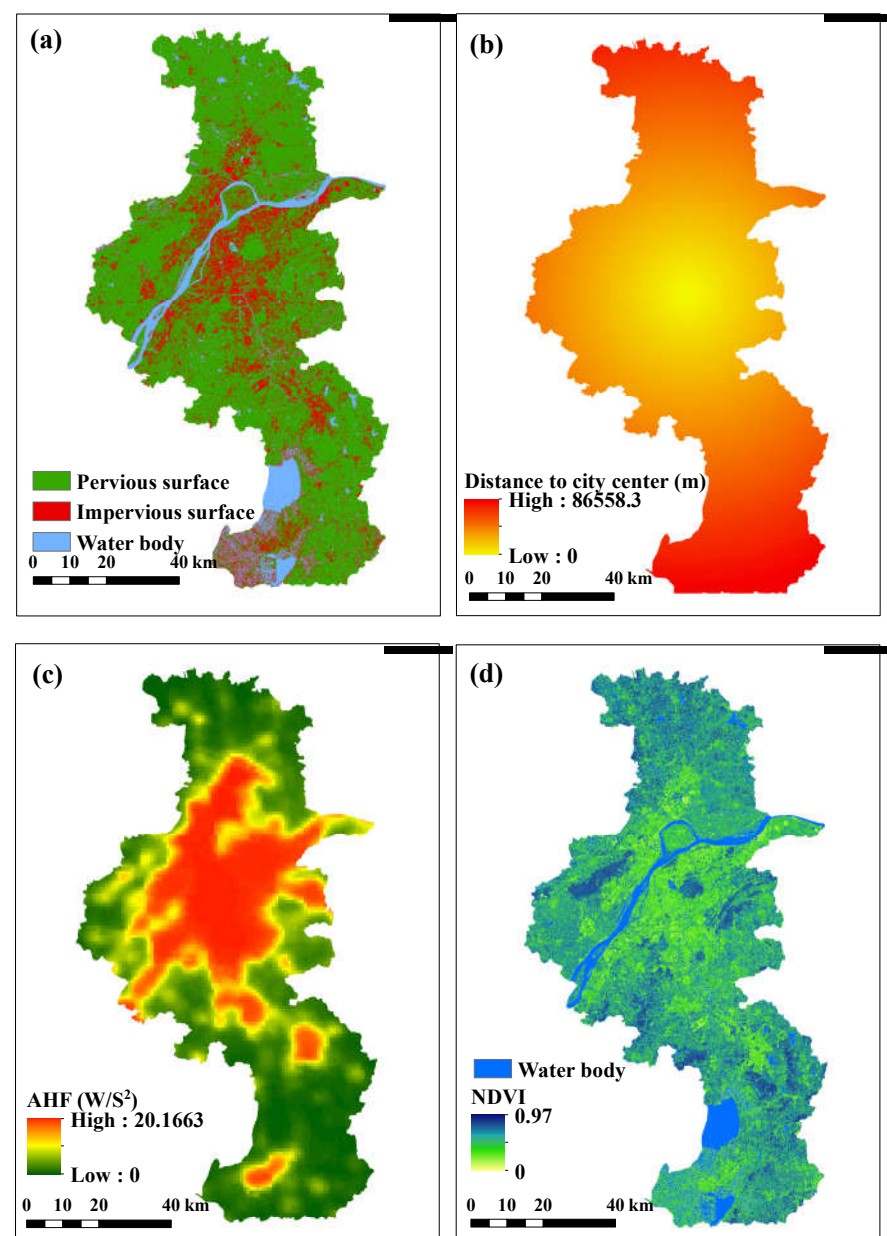

**Figure 2: Spatial distribution of some environmental variables on 21 July 2017 in Nanjing city: (a) impervious surface; (b) distance from city center; (c) AHF (anthropogenic heat flux); (d) NDVI (normalized difference vegetation index).**






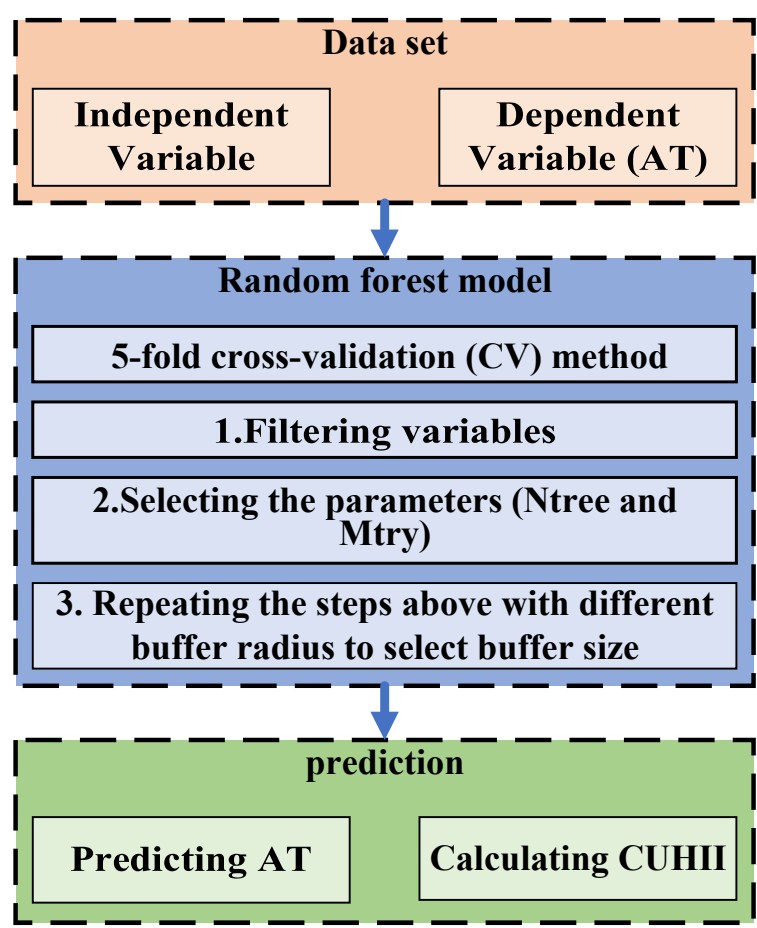

**Figure 3: Flowchart for constructing the RF model and evaluating the CUHII (Canopy urban layer heat island intensity). Abbreviations: DEM, digital elevation model; AHF, anthropogenic heat flux; LULC, land use/land cover; IS, impervious surface; AT, air temperature; RF, Random Forest.**



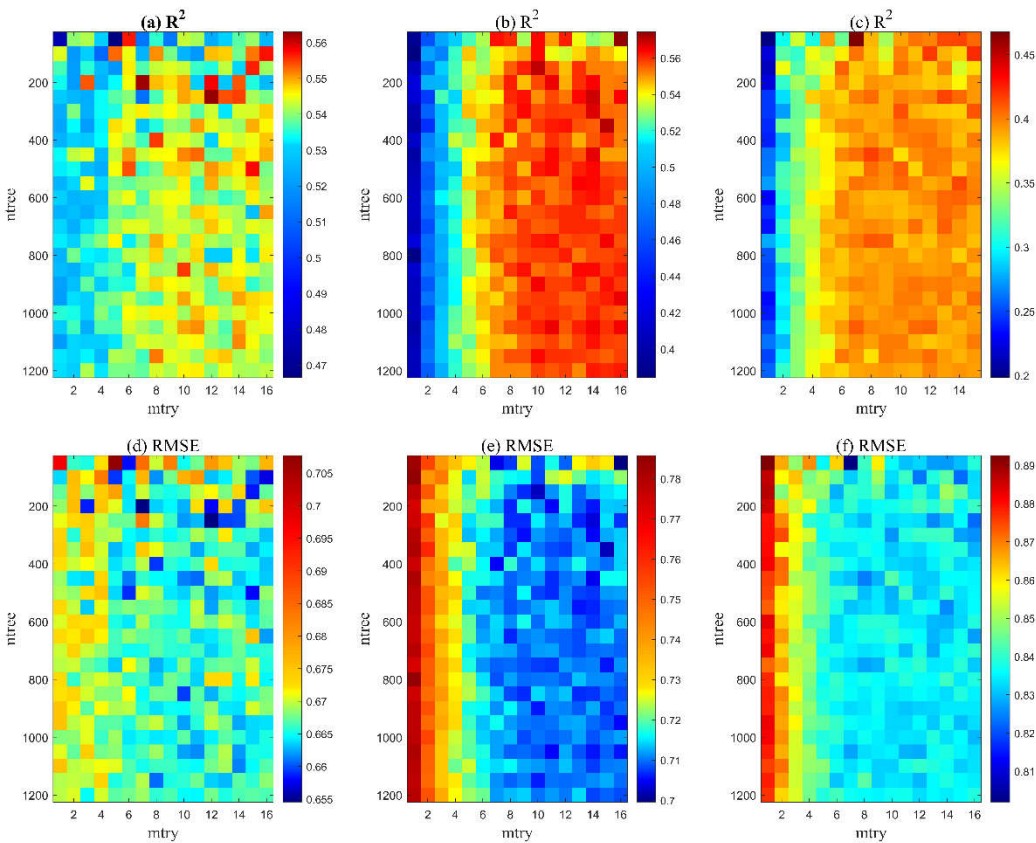

**Figure 4.** The (a–c) $R^2$ (coefficient of determination) and (d–f) RMSE (root-mean-square error) changes with the parameters Ntree and Mtry of the model using the dataset on (a, d) 11 August 2013, (b, e) 2 September 2015, and (c, f) 21 July 2017.






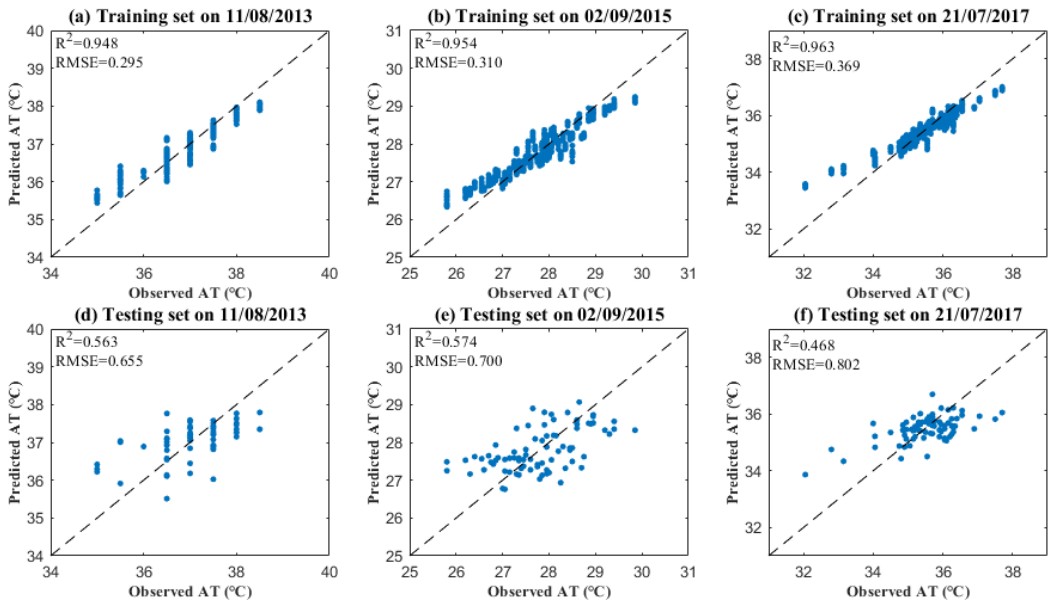

**Figure 5: Scatterplot of predicted and observed air temperature: five-fold cross-validation (CV) for the training set on (a) 11 August 2013, (b) 2 September 2015 and (c) 21 July 2017; five-fold CV for the testing set on (d) 11 August 2013, (e) 2 September 2015 and (f) 21 July 2017.**




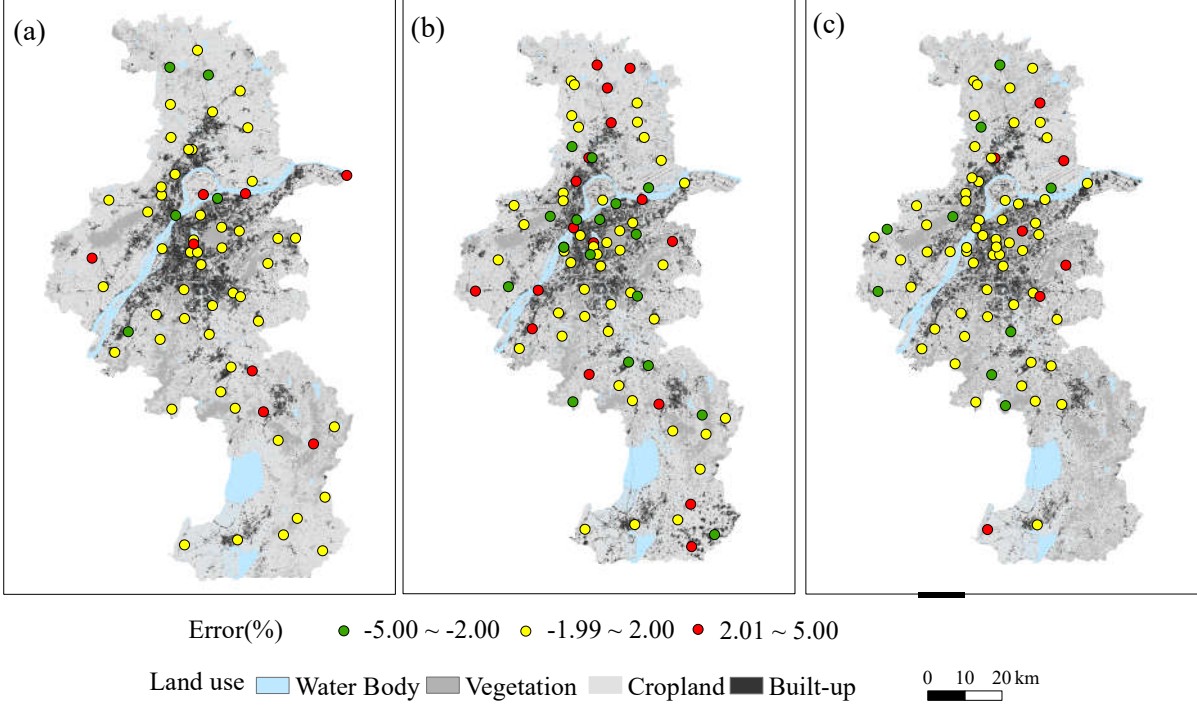

Figure 6: the predicted relative error of the air temperature by random forest: (a) 11 August 2013; (b) 2 September 2015; (c) 21 July 2017.



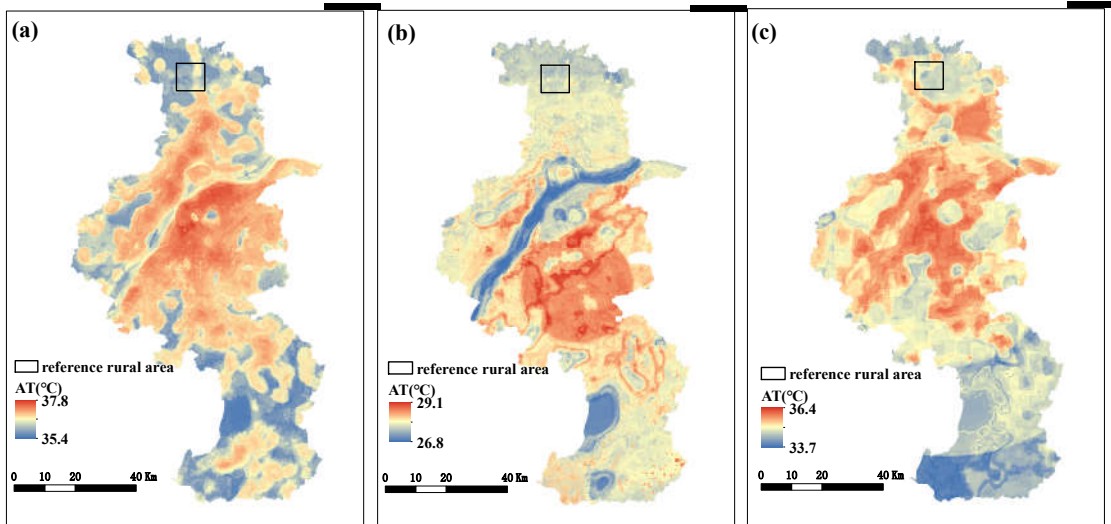

**Figure 7: Spatial distribution of air temperature (AT) in Nanjing and the reference rural area: (a) 11 August 2013; (b) 2 September 2015; (c) 21 July 2017.**



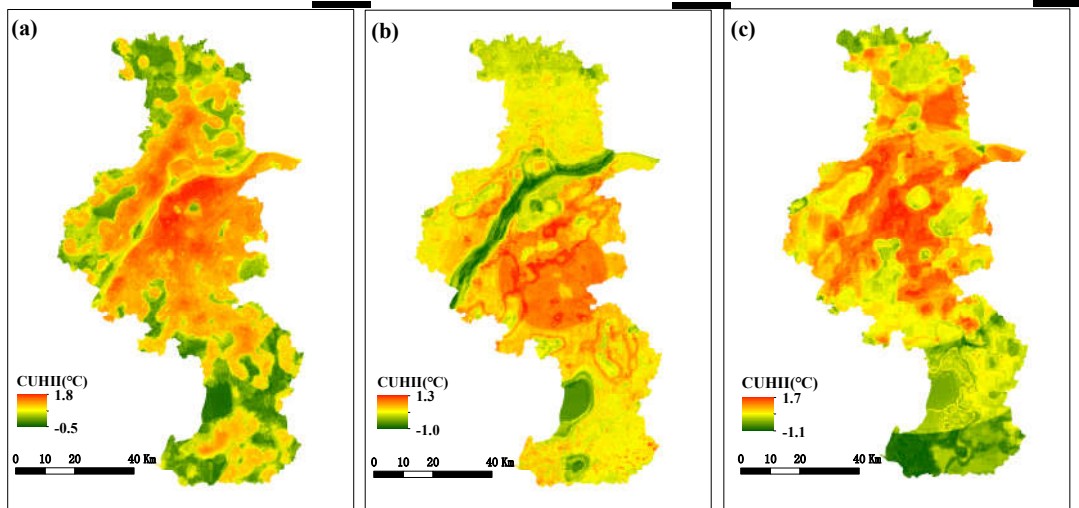


**Figure 8: Spatial distribution of the CUHII in Nanjing: (a) 11 August 2013; (b) 2 September 2015; (c) 21 July 2017.**


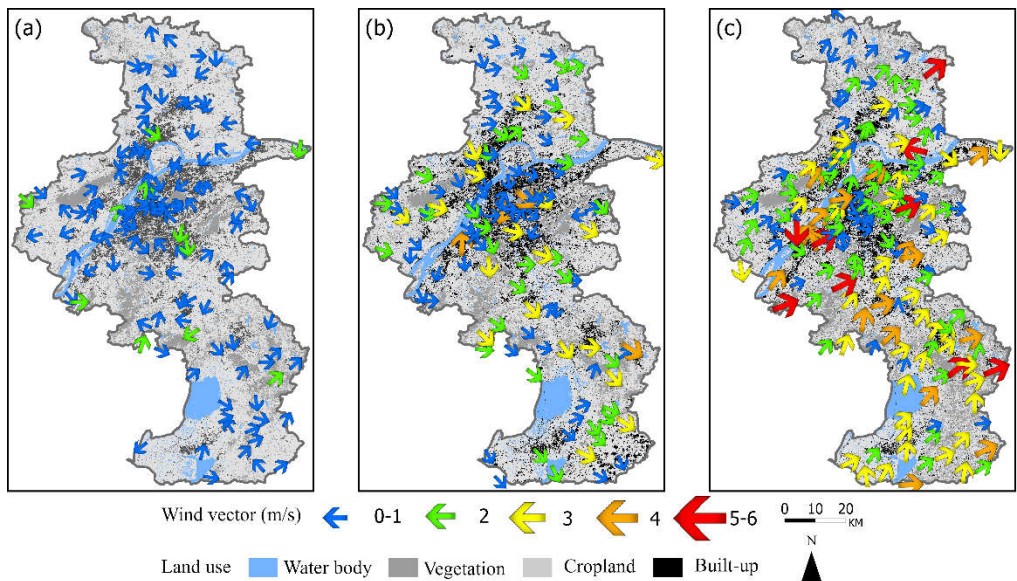

**Figure 9: Wind vector field in Nanjing on (a) 11 August 2013, (b) 2 September 2015, and (c) 21 July 2017.**






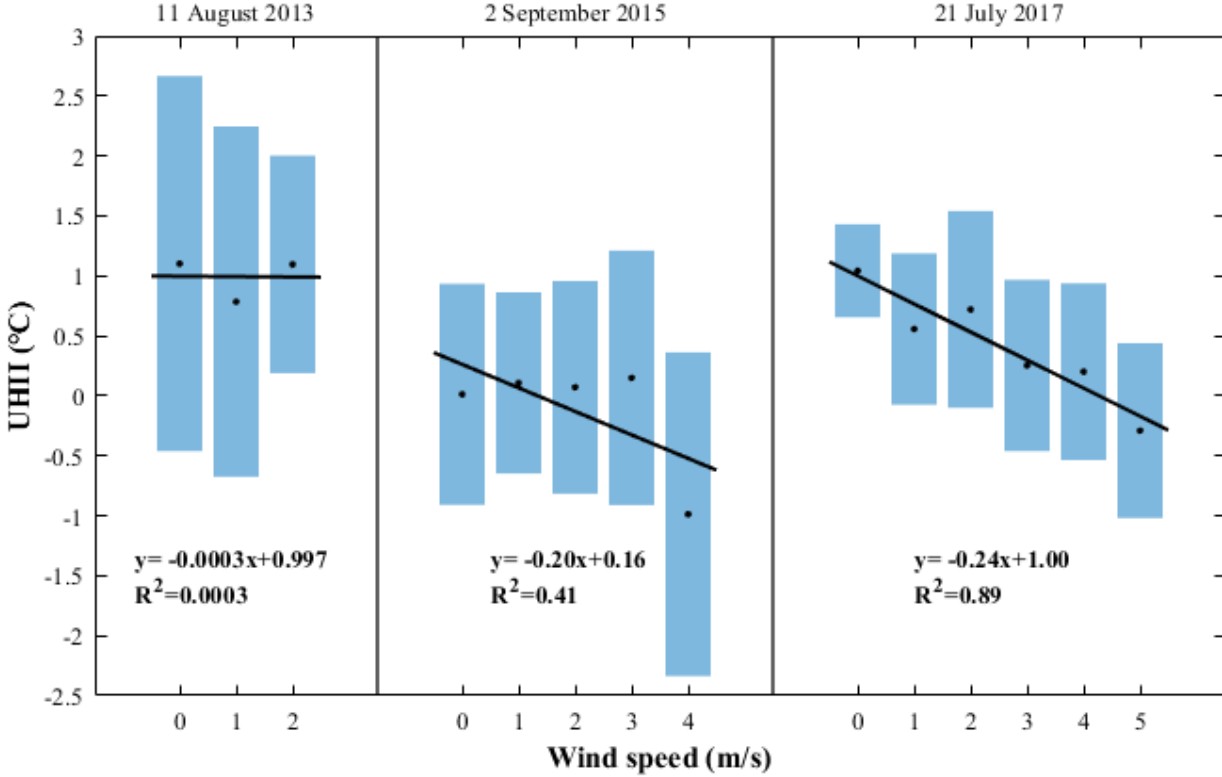

**Figure 10:** Relationship between urban heat island intensity (UHII) and wind speed around all meteorological stations on (a) 11 August 2013, (b) 2 September 2015, and (c) 21 July 2017. The black dots represent the mean UHII, while the bottom and top edges of the box indicate the uncertainties of one standard value from the mean






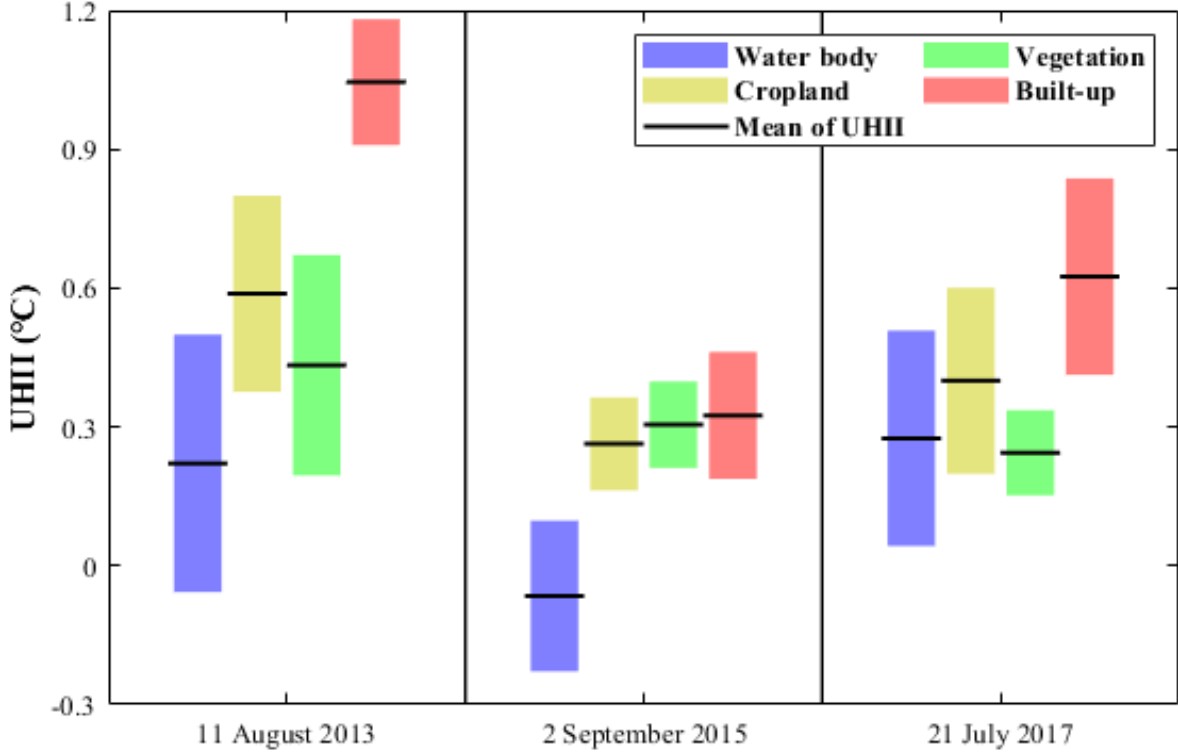

**Figure 11: Histogram of urban heat island intensity (UHII) over different land use/land cover on (a) 11 August 2013, (b) 2 September 2015, and (c) 21 July 2017. Horizontal lines represent the mean intensity, while shaded regions are the uncertainties of one standard value from the mean.**






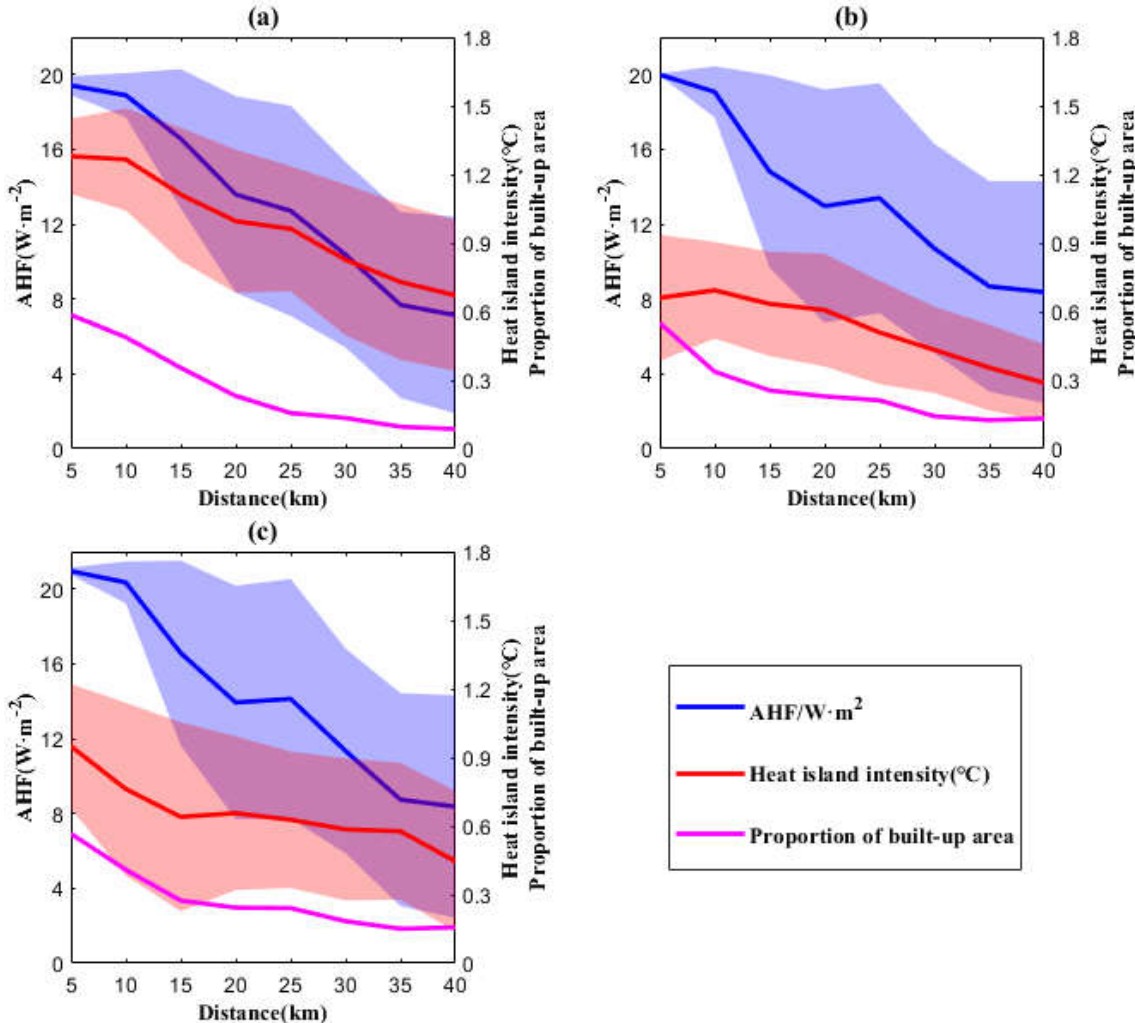

**Figure 12: Changes in air temperature, anthropogenic heat flux (AHF) and the proportion of built-up areas with distance from the city center on (a) 11 August 2013, (b) 2 September 2015, and (c) 21 July 2017. Thick lines represent mean values, while shaded regions are the uncertainties of one standard value from the mean.**






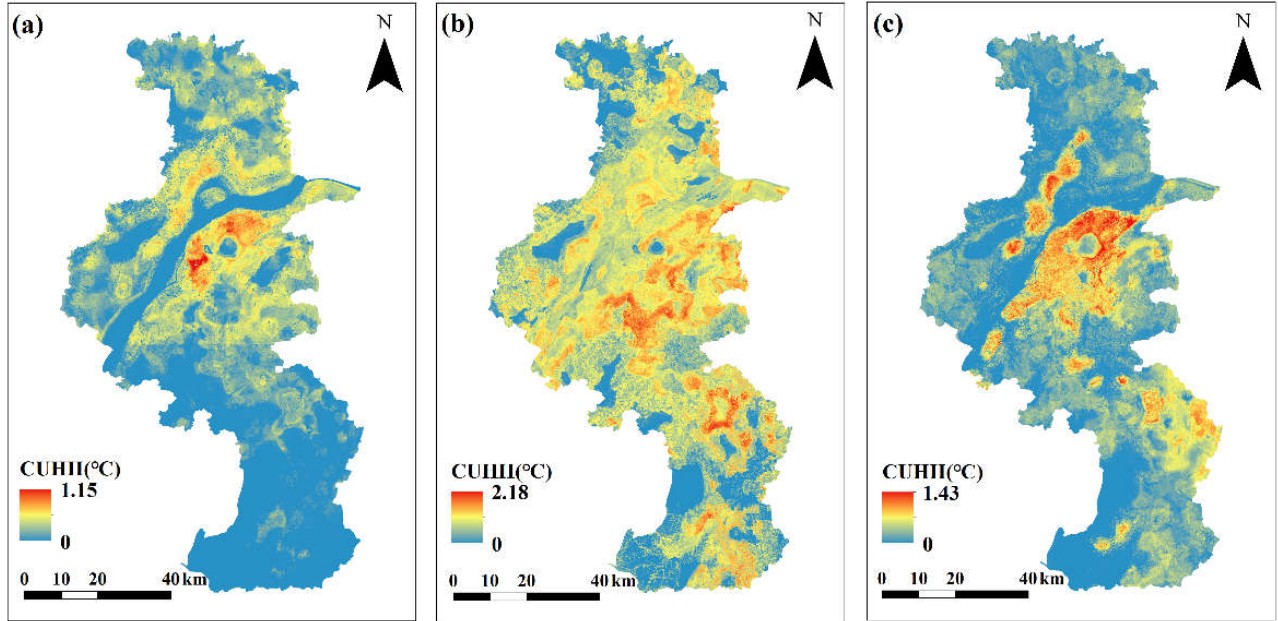

**Figure 13: Spatial distribution of canopy urban heat island intensity (CUHII) in Nanjing during a heat wave period: (a) 12 August 2013; (b) 13 August 2013; (c) 14 August 2013.**