# Peer review of "A High-Resolution Monitoring Approach of Canopy Urban Heat Island using Random Forest Model and Multi-platform Observations"

_Atmospheric Measurement Techniques, 2021_

## Author Comment (AC1)

**Response to reviewer comments**

We are sincerely grateful to editor and reviewers for their valuable time spent on reviewing our manuscript. The comments are very helpful and valuable, and we have addressed some issues raised by the reviewers in the revised manuscript. Please find our point-by-point response (in blue font) to the comments (in black font) raised by reviewers.

**Reviewer 2**

In the manuscript the authors proposed a framework to train the random forest (RF) model so as to output high resolution air temperature map. The RF model takes various environment parameters (mostly derived from remote sensing data) around a dense network of meteorology stations as independent variables and it is trained using the air temperature data observed by those stations. The work starts with a good motivation to provide high resolution air temperature map, which is important for a better understanding of thermal environment of the city and helps for heat exposure assessment.

The method propose by the authors shows the capability of generating high resolution air temperature map. With this temperature map the authors further studied the fine scale canopy layer urban heat island as well as its relationship with driving factors. This, from my point of view, matches well the original motivation of this work.

The workload of this study involves quite a bit effort, especially regarding the preparation of the input variables that are fed to the RF model, and the identification of the proper buffer size, as well as the comparison of the model performances between the proposed RF model and other regression models.

As I did not really give a fast review before the discussion started, in the 'fast review' I raised several questions, such as increasing the presentation quality, better organizing the technical stuff, giving more space to technique details, I appreciate the authors' great effort in address my concerns, I found they have been addressed mostly. So I personally do not see fundamental issues in the current version of the manuscript.

**Response:** Many thanks for your positive and valuable comments, and kind suggestions in both quick review and current rounds. We hope our revisions have properly addressed the various concerns and issues raised.

There are a few minor issues remaining:
L34: "relatively" is unnecessary.
**Response:** Amended.

L83: I suggest replacing 'Meanwhile' with 'However', and moving this passage to the following paragraph.
**Response:** Amended.

L104-105: I find the last sentence is confusing and is not necessary to put it here.
**Response:** Many thanks for your kind suggestion. We have deleted the sentence.

L107: a link directing to 'geospatial data cloud' would be fairer.

**Response:** Many thanks for your kind suggestion. The link is added.

L269-277: Please be aware that due to different weather and climate conditions, it is very risky to directly compare the UHI intensities, even though you seem to have carefully selected the same barely impacted rural reference area. The rural area and urban area respond to different parameters very differently. If you really want to compare UHI intensities from different, I would suggest classifying the UHI intensities into several levels using the quantile instead of the uniform scalar values. If not so, please at least avoid direct comparison, while just plainly describing the table content is fine.

**Response:** Thank you very much for the suggestion. We deleted the comparison part and modified this paragraph.

L406: I would suggest putting it this way: "the RF prediction framework proposed in this work not only can dynamically predict CUHII in detail within highly heterogeneous cities, but also can be built….". Besides, I see the potential that the model can be used cross a short period as most of the environmental parameters fed to the model probably can remain stable for some time, but I would say that it is rather bold to claim a period like one month or even longer.

**Response:** Thank you very much for the constructive suggestions. We have modified the sentence. The random forest models with the hourly air temperature data were modeled in August 2013, September 2015 and July 2017, and figure 7 shows the model accuracy is higher by using more data in one month with respect to one day (Figure 5). Thank you again for help us improve our model. The above part has been added in Lines 240 to 249 in the revised manuscript. In addition, we also discussed the potential of the model at time scales at lines 452-454.

[Figure]

Figure 7: Scatterplot of predicted and observed air temperature using data in a month five-fold CV for the testing set on (a) August 2013, (b) September 2015 and (c) July 2017.

L733: in the caption of figure3, it should be 'canopy layer urban heat island' instead. Please note that in the figure you no longer have Abbreviations like DEM, AHF, LULC, etc.

**Response:** Many thanks for your kind suggestion. Amended.

L764. In the caption of figure10, I suppose by 'standard value', you mean standard deviation.

**Response:** Thanks for your suggestion. Amended.

---

## Author Comment (AC2)

**Response to reviewer comments**

We are sincerely grateful to editor and reviewers for their valuable time spent on reviewing our manuscript. The comments are very helpful and valuable, and we have addressed some issues raised by the reviewers in the revised manuscript. Please find our point-by-point response (in blue font) to the comments (in black font) raised by reviewers.

**Reviewer 3**

This study aims to integrate remote sensing data with ground-based temperature measurement to predict canopy heat island at a large spatial extent. Overall the idea is interesting, but the data size is very limited and the method seems to be a typical approach. Below are my major comments:

**Response:** Many thanks for your positive comments. We are very grateful for all the constructive comments and suggestions. We have adopted all the suggestions in our revised manuscript.

1) Line 55, "meteorological measurements and high-density observations", it is not clear what the authors are referring to here. Does it mean rural station against microclimate observations? Please rephrase here.

**Response:** Thanks for your kind suggestion. Rephrased as follow:
In-situ (field) measurements include conventional measurements from sparse national meteorological stations, and high-density microclimate observations from experiments or high-density automatic sites over various underlying surfaces.

2) Line 104, ""Nanjing's UHI was observed to be 0.5 in 2005". It is important to clarify how the 0.5 is calculated, as it does not contain any spatial variation, and differs from the results in this study.

**Response:** Thank you for your comments. The previous reference employed the observation data of three weather stations located in the urban area of Nanjing and that of one weather station in rural areas to explore UHI in Nanjing. As Nanjing has been developing rapidly from 2005, the built-up area and population are increasing rapidly, so the canopy urban heat island intensity was increasing accordingly, so our results are higher. For readerships, we have deleted this sentence.

3) Line 110, my major concern about this study is the usage of only 3 snapshots of the satellite measurement. Only 3 days are selected over a 5-yr period from 2013 to 2017. Such data availability is surprising low. On top of this, this performance of RF model is only marginal, with R2 about 0.5 in the cross-validation. How would the authors justify the potential or accuracy of the model in predicting the urban heat island for practical usage, if the model is to be extended to more days under complex weather conditions?

**Response:** Thank you very much for your constructive comments. the present three snapshots is under clear sky conditions, which is favorable to retrieve the stable input factors such as NDVI, surface albedo, land use, etc, which are similar in one month or even in a season. For more snapshots, the cloud and air pollution effects on retrievals should be amended in the future. The aim of our present work is to propose a refined assessment framework of CUHII was then established by using Random Forest Model with observed AT and environmental variables as a demonstration.

To improve the practicality of the model, we added the discussion of model robustness in line 240 to 249. The random forest models with the hourly air temperature data were modeled in the whole August 2013, September 2015 and July 2017, and figure 7 shows the model accuracy is higher by using more data in one month with respect to one day (Figure 5). Thank you again for help us improve our model. In addition, we also discussed the potential of the model at time scales at lines 452-454.

[Figure]

Figure 7: Scatterplot of predicted and observed air temperature using data in a month five-fold CV for the testing set on (a) August 2013, (b) September 2015 and (c) July 2017.

4) Line 113, what does 0.5 intervals mean?

**Response:** Thanks for your suggestion. 0.5 ℃ interval means that the resolution of temperature measurement. We have changed it as following: "with resolution of $0.5℃$ on 11 August 2013 and $0.1℃$ on 2 September 2015 and 21 July 2017"

5) Line 124, Does the AHF data vary diurnally and seasonally? If it does not, then the AHF data may be less meaningful to be incorporated into the RF model. Instead of the LULC map, I will suggest authors to add the AHF map here.

**Response:** Thank you very much for your kind suggestions. The AHF here varied annually. We expect that AHF distribution can shape the main morphology of urban thermal environment. We can not get AHF data at diurnal and seasonal scales. In future, if we got high-temporal-resolution AHF data, we will update them in the model. We have point this limit in lines128-130. In Figure 1, we have changed the LULC map to the AHF map.

6) Line 150, a table summarizing the predictors with their sources and resolution used

in the RF model will be very useful.

**Response:** Thank you very much for your kind suggestion, we have added table as following.

**Table 1: Independent variables with their sources and spatial resolution**

|  | Parameters | Source | Spatial resolution (m) |
|---|---|---|---|
| geometric parameters | Proportion of LULC area | Landsat 8 data | 30 |
|  | Latitude and longitude distance from the city center | LULC data |  |
|  | Altitude, Slope and aspect | DEM data |  |
| physical parameters | Proportion of IS area
Albedo
NDVI, NDBI, gNDVI, SAVI and NDMI | Landsat 8 data | 30 |
| anthropogenic parameters | AHF data | NOAA nighttime lighting data | 1000 |

Notes: DEM, digital elevation model; IS, impervious surface; NDVI, normalized difference vegetation index; NDBI , normalized difference built-up index; gNDVI, green normalized difference vegetation index; SAVI, soil-adjusted vegetation index; NDMI, normalized difference moisture index; AHF, anthropogenic heat flux.

7) Line 180, Different variables are used on different days, and the optimal Mtry and Ntree also change substantially. This essentially means the built model can only apply for a specific day. Given that the study only focus on Nanjing under clear sky conditions, I am concerned about the applicability of the model. If we utilize this method to study CUHI in the future, that means one will need to run it for every hour, and the accuracy is not guaranteed even under the clear sky condition. What I suggest may be a lot of work, but I think using more models and comparing their performance and come up with a consistently well-perform model is really needed to enhance this paper to a higher quality/level.

**Response:** Thank you very much for your constructive suggestion. To improve the applicability, we added the discussion of model robustness in lines 235-247 of the revised manuscript which was mentioned in response 3.
We have tried to compare the accuracy of RF model with that of stepwise linear regression and GWR in lines 204- 209 and Table 2. However, stepwise linear regression cannot deal with so much data (44375, 51673 and 53973 samples in three times). Such a large matrix leads to a huge burden on calculations and memory, while RF model can easily avoid the problem by processing samples one by one. Neither stepwise linear regression nor GWR can fit complex data, such as periodicity in time

series data in this study.

In future, we would like to compare different machine learning methods to come up with a consistently well-perform model, e.g., SVM and ANN. We will also use stacking ensemble strategy to combine the advantages of different models and get the best prediction results. We have added this perspective in lines 430-432.

8) Line 197, from my perspective, using a buffer size of 2-km to predict temperature at 30-m resolution is not scientifically sound. As expected, the estimated spatial variability of air temperature/CUHI is small. This contradicts the local climate zone framework that local urban landscape may dominate the air temperature under a similar weather condition. Did the authors check the spatial map of SUHI? I believe the spatial variability of LST will be much larger. I suggest authors add a map of LST and compare it with the estimated CUHI. That can help facilitate the discussions on your model.

**Response:** Thanks for your suggestions. We have compared the accuracy based on different buffer size in line 195 to 203 of the revised manuscript, showing the best performance at 2000-m buffer size. We also added the LST map of Nanjing in section 4.1 to compare air temperature map with LST map, showing similar distribution at light wind conditions while different distribution in large wind conditions. This is because satellite-derived LST distribution mainly depended on the longwave emissivity related to LULC distribution, while air temperature is not determined by LST but other local meteorological factors (e.g., wind and humidity).

9) Figure 6 and Figure S2, why is the number of stations different in three subplots?
**Response:** Thanks for your suggestion. Because of the increase and decrease of high-density automatic stations every year, the number and locations are different.

10) Line 406. It is good that authors summarize the existing approach of predicting AT in the literature. Given that RF model is a very typical approach adopted by many studies, what is the novelty of this study? Maybe the authors can elaborate more here.
**Response:** Thank you for your kind suggestions. We elaborate more novelty in lines 445 -454 of the revised manuscript as followings:
"The RF prediction framework proposed in this work can not only can dynamically predict CUHII in detail and high frequency within highly heterogeneous cities, but also can be built against different weather backgrounds, mainly because the environmental parameters entered into the model are relatively stable within a certain period (such as the same month or season). As long as the environmental parameters are acquired once, they can be combined with the AT data in real time to establish the RF model, and the spatial distribution characteristics of CUHII with high temporal and spatial resolution can be obtained. For instance, we randomly predicted the 30-m-resolution AT and spatial distribution of CUHII (Figure 15) with the wind vector field (Figure S3) during the heat wave period of 12–14 August 2012, thereby supporting those involved in making decisions with respect to urban climate, urban

planning and urban energy consumption. Particularly, the potential that our proposed model can be used cross a short period as most of the environmental parameters fed to the model probably can remain stable for some time, e.g., one month or even longer. "